# Structural dynamics of myosin 5 during processive motion revealed by interferometric scattering microscopy

Joanna Andrecka[1†], Jaime Ortega Arroyo[1†], Yasuharu Takagi[2], Gabrielle de Wit[1], Adam Fineberg[1], Lachlan MacKinnon[1], Gavin Young[1], James R Sellers[2], Philipp Kukura[1*]

[1]Physical and Theoretical Chemistry Laboratory, Department of Chemistry, University of Oxford, Oxford, United Kingdom; [2]Laboratory of Molecular Physiology, National Heart, Lung and Blood Institute, National Institutes of Health, Bethesda, United States

**Abstract** Myosin 5a is a dual-headed molecular motor that transports cargo along actin filaments. By following the motion of individual heads with interferometric scattering microscopy at nm spatial and ms temporal precision we found that the detached head occupies a loosely fixed position to one side of actin from which it rebinds in a controlled manner while executing a step. Improving the spatial precision to the sub-nm regime provided evidence for an ångstrom-level structural transition in the motor domain associated with the power stroke. Simultaneous tracking of both heads revealed that consecutive steps follow identical paths to the same side of actin in a compass-like spinning motion demonstrating a symmetrical walking pattern. These results visualize many of the critical unknown aspects of the stepping mechanism of myosin 5 including head–head coordination, the origin of lever-arm motion and the spatiotemporal dynamics of the translocating head during individual steps.

*For correspondence: philipp. kukura@chem.ox.ac.uk

†These authors contributed equally to this work

Competing interests: The authors declare that no competing interests exist.

## Introduction

Myosin 5a moves in a hand-over-hand fashion with individual heads moving by 74 nm and the center of mass by 37 nm for each step (*Vale, 2003*; *Sellers and Veigel, 2006*; *Hammer and Sellers, 2012*). A finely tuned kinetic sequence of ATP binding, hydrolysis, phosphate release and eventually ADP release by the enzymatic heads enables processive motion towards the '+' end of actin filaments (*De La Cruz et al., 1999*; *De La Cruz and Ostap, 2004*; *Rosenfeld and Sweeney, 2004*; *Sakamoto et al., 2008*; *Forgacs et al., 2008*). One of the most remarkable aspects of myosin 5a is the efficient conversion of chemical energy by the molecular-sized enzymatic heads into a translation of 74 nm along a 7 nm diameter actin filament in the presence of Brownian motion and a crowded cellular environment. Crystal structures of myosin 5a exist either with or without nucleotide, but only in the detached state (*Coureux et al., 2003*, *2004*). Although these structures served as a model for the structural changes potentially induced by binding to actin, (*Volkmann et al., 2005*) the internally coupled rearrangement of the subdomains leading to ADP and phosphate release or dissociation from actin upon ATP binding could only be studied in silico (*Cecchini et al., 2008*, *2010*; *Sweeney and Houdusse, 2010*; *Preller and Holmes, 2013*). The experimental challenge in revealing such structural dynamics has been largely of spatiotemporal origin, since they are likely on the Ångstrom scale and may be very transient.

Similarly, there has been considerable interest in directly revealing the motion of the individual heads as myosin steps along actin in an attempt to unravel the origins of efficient motion on the nanoscale. Single molecule studies employing optical trapping (*Mehta et al., 1999*; *Rief et al., 2000*)

**eLife digest** Cells use motor proteins that to move organelles and other cargos from one place to another. The myosins are a family of motor proteins that pull cargo along filaments made of another protein called actin. The 'head' end of myosin attaches to the actin filament and the 'tail' end binds to the cargo. The head and tail are connected by a flexible linker that allows the protein to change shape.

The tails of two myosin molecules bind together to form a two-headed motor that can move along an actin filament by taking 74 nm steps. At the start of each step, both heads are attached to the actin with one in front of the other. The leading head remains attached while the rear head detaches from actin and moves in front of the leading head reattaching to actin. These movements are repeated many times to allow the motor to move along the filament, much like a tightrope walker walking along a wire. However, it is not known how the motor can move so efficiently along the actin while managing to avoid falling off its track.

Here, Andrecka et al. analyzed the movement of the myosin 5 motor in real time using a method called 'interferometric scattering microscopy'. The experiments show that when a head detaches from the actin, it is temporarily held out to one side of the actin filament. From here, this detached head sways back and forth, until it takes a step forward and binds firmly to the next position on the filament. Both heads follow identical paths along the actin filament, and so the movement resembles the way that a drawing or dividing compass can be used to measure distances on a map.

Andrecka et al.'s findings shed new light on how myosin motors move along actin. This may become a blueprint for efficient nano-mechanical motion and could therefore be important for designing artificial machines that operate on the nanoscale.

and fluorescence imaging (*Forkey et al., 2003*; *Yildiz et al., 2003*; *Snyder et al., 2004*; *Warshaw et al., 2005*) have either reported periods of increased flexibility, (*Veigel et al., 2002*; *Dunn and Spudich, 2007*; *Beausang et al., 2013*) or partitioning of the step into sub-events (*Veigel et al., 2002*; *Uemura et al., 2004*; *Cappello et al., 2007*; *Sellers and Veigel, 2010*). All resulting models involve a forward aiming power stroke followed by a Brownian search mechanism (*Veigel et al., 2002*; *Okada et al., 2007*; *Shiroguchi and Kinosita, 2007*; *Karagiannis et al., 2014*). The power stroke of the attached head contributes only partially to the step by moving the pivot point that facilitates Brownian rotation of the unbound head. The bias towards the next binding site is believed to be provided by the recovery stroke and its stability, but how unidirectional stepping is achieved in the presence of such a large and mostly random motion remains unclear (*Shiroguchi et al., 2011*). The first passage time of the translocating head is expected to be on the order of 100 µs, (*Veigel et al., 2002*; *Hinczewski et al., 2013*) although the head spends tens of ms in the unbound state given the maximum rates of ATP hydrolysis and binding to actin (*Veigel et al., 2002*; *Dunn and Spudich, 2007*; *Beausang et al., 2013*).

Any technique aiming to visualize the structural dynamics of the motor domain or the motion of the unattached head would thus have to achieve simultaneous millisecond temporal and nanometer spatial precision. Optical trapping operates in this spatiotemporal regime, but it is not possible to monitor the motion of the unattached head without significant perturbation. Smaller labels affixed to the myosin motor domain or to calmodulin molecules on the lever arm, as available in fluorescence imaging, do not suffer from this limitation, but cannot provide sufficient localization precision on the millisecond time scale. We therefore designed an assay based on interferometric scattering microscopy (iSCAT), (*Kukura et al., 2009*; *Ortega-Arroyo and Kukura, 2012*) a technique that has recently been shown to enable simultaneous high-speed and high-precision imaging of 20 nm diameter or smaller nanoscale scattering labels (*Andrecka et al., 2013*; *Ortega Arroyo et al., 2014*).

## Results

### Individual steps proceed via a single, spatially-constrained transient state of the detached head

We attached a 20 nm gold label functioning as an efficient light scatterer to the N-terminus of the myosin 5 head and tracked its motion as the motor travels along actin filaments (*Figure 1A*, inset).

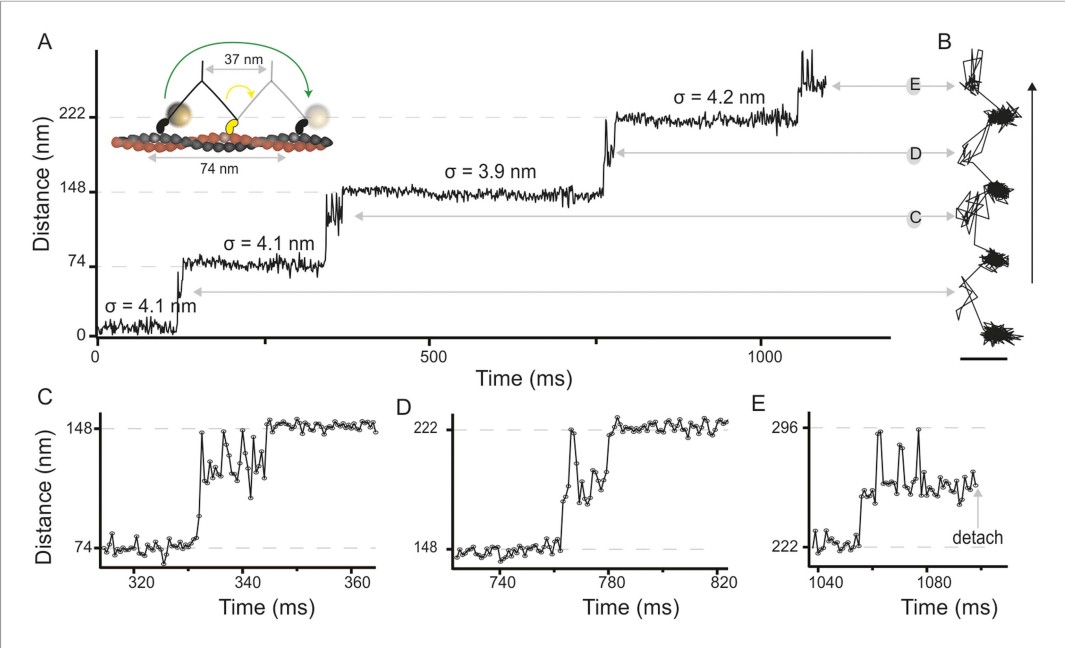

**Figure 1**. High-speed nanometric tracking of myosin 5 with interferometric scattering (iSCAT) microscopy. (**A**) Distance traveled as a function of time for a single myosin 5 molecule biotinylated at the N-terminus and labeled with a 20 nm streptavidin-functionalized gold particle. The lateral localisation precision, σ, defined as the standard deviation of the positional fluctuations of the label while bound to actin is given above each of the actin-attached periods. Inset: schematic of gold-labeled myosin 5 stepping along actin. (**B**) Corresponding 2D-trajectory with the arrow indicating the direction of movement. (**C–E**) Close-up of the transient states indicated in **A** and **B**. ATP concentration: 10 µM. Scale bar: 50 nm. Imaging speed: 1000 frames/s (corresponding video: *Video 1*).

The following figure supplements are available for figure 1:

**Figure supplement 1**. Activity of myosin 5 labeled with 20 nm gold nanoparticles at the N-terminus.

**Figure supplement 2**. Detection of the transient state with a molecular sized fluorescent label.

**Figure supplement 3**. Rare transient unbinding events for the leading head of myosin 5.

From the centre of mass of the signal produced by the label, we determined the position of the head as a function of time revealing discrete 74 nm steps at 1000 frames/s imaging speed with ~4 nm positional precision (*Figure 1A*, *Figure 1—figure supplement 1A*, *Video 1*). We verified that the addition of the 20 nm label did not interfere with the mechano–chemical cycle of myosin 5a by characterizing the speed of movement at different ATP concentrations. Our results compare well with those from a series of other single molecule studies where the molecule was labeled on the lever arm, the tail, or attached to a surface (*Figure 1—figure supplement 1B*).

In addition to the expected 74 nm steps, we observed periods of increased positional fluctuations between detachment and reattachment of the labeled head, previously interpreted as a signature of Brownian search (*Dunn and Spudich, 2007*). Our effective lateral localization precision of 4 nm at 1000 frames/s, however, enabled us to visualize the motion of the unbound head precisely and revealed a transient state with a center of mass just over half way between the two binding sites and offset by 40 nm perpendicular to the actin filament (*Figure 1B*). From this position, the myosin head repeatedly moved back and forth between the next actin binding site and the transient state position (*Figure 1C,D*). The same behavior sometimes occurred just before myosin detached from the filament at the end of a trajectory (*Figure 1E*), although detachment of myosin with the labeled head bound was equally likely. On the rare occasions that the labelled leading head of myosin detached and reattached, it occupied a similar position in space to a translocating head (*Figure 1—figure*

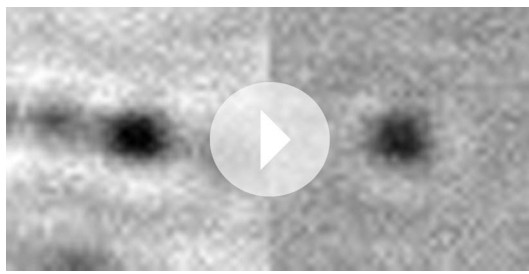

**Video 1.** This file contains the iSCAT video from which the trajectory shown in *Figure 1* of the main manuscript was obtained. The left panel presents the raw video format after referencing with the flat field image. The flat field accounts for illumination inhomogeneities and was generated as described in the Materials and Methods section. The right panel shows the static background subtracted video format. The background includes the average of 100 frames before the particle binds and removes the constant features which are part of the sample (a.g. actin, glass roughness) by simple subtraction (*Ortega Arroyo et al., 2014*). The video was taken at 1000 frames/s and at a magnification of 31.8 nm/pixel. The corresponding field of view shown is 1.30 × 1.30 μm² and corresponds to a total recording time of 1.3 s. The video playback speed is 1000 frames/s.

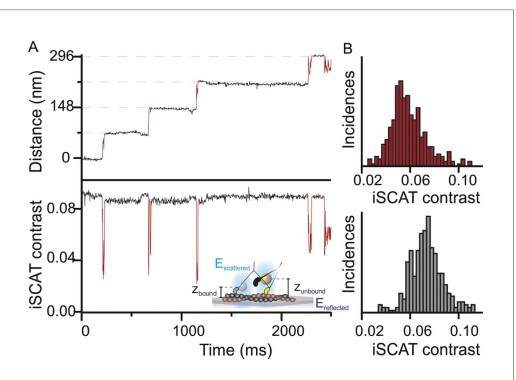

**Figure 2**. Three-dimensional interferometric tracking of the myosin head. (**A**) Distance trace (upper panel) with the simultaneously recorded iSCAT contrast (lower panel). The red subset of the traces corresponds to the unbound head state. Inset: schematic illustrating how the difference in optical path difference for the bound ($z_{bound}$) and unbound ($z_{unbound}$) state leads to changes in iSCAT contrast caused by the interference of the reflected ($E_{reflected}$) and scattered ($E_{scattered}$) electric fields. (**B**) Normalized histogram of the average iSCAT contrast while the head is in the transient state (red, N = 329) or bound to actin (black, N = 303). ATP concentration: 10 μM. Imaging speed: 1000 frames/s.
The following figure supplement is available for figure 2:

**Figure supplement 1**. Dual colour iSCAT imaging of myosin 5.

supplement 3), suggesting that the transient state represents a true potential minimum of the one head bound state of myosin.

## Revealing the unbound head motion in three dimensions

In addition to providing nanometer precise information on the lateral position of the head through the center of mass of the signal, the iSCAT signal magnitude is very sensitive to the label-to-surface distance allowing for overall nanometric localization in three-dimensions (*Krishnan et al., 2010*). This property arises from the interferometric nature of the technique, which scales the iSCAT signal as $sin\Phi$, where $\Phi$ is the phase difference between scattered and reflected light fields. As the label moves perpendicular to the sample plane, the optical path length and with it the phase difference between the two fields changes by the following relation $\Phi = 4\pi\eta z/\lambda$ (*Figure 2A*, inset). A complete signal inversion occurs for a displacement of $z = \lambda/4\eta = 84$ nm, where $\lambda$ is the illumination wavelength (445 nm) and $\eta$ the refractive index of the medium (1.33). Thus, variations in the axial distance of the motor domain with respect to the filament bound to the sample surface lead to changes in the scattering contrast. We often observed these changes in contrast between the actin bound and unbound states (*Figure 2A*) and obtained an average change in iSCAT signal during the step (*Figure 2B*). The drop of 3.5% in iSCAT contrast (~40% of the total signal) suggests that the myosin head lifts on average by 24 ± 10 nm from its actin bound position. Although the precision of the measurement was in principle higher, we could not accurately determine the additional phase contributions to the interferometric signal on an individual label basis, which are required for a robust calibration.

We occasionally encountered tracks where the transient state could not be localized because the iSCAT contrast dropped to zero. We interpret these cases as corresponding to molecules oriented parallel to the glass surface based on simultaneous iSCAT measurements using blue (445 nm) and red (635 nm) illumination. The particle remained visible in the red channel while it disappeared in the blue. At the same time, the off-axis component of the transient state was much smaller compared to those trajectories when the transient state remained visible in both channels, which is most consistent with a parallel bound molecule lifting the detached head up by ~40 nm (*Figure 2—figure supplement 1B*).

## The transient states for both heads are located on the same side of the actin filament

To investigate whether subsequent steps occur on the same or opposite sides of the actin filament, commonly referred to as symmetric and asymmetric hand-over-hand stepping, (*Hua et al., 2002*) we labeled the two heads of a single molecule differently, one with a quantum dot and the other with a 20 nm gold particle (*Figure 3*, top). Simultaneous detection of quantum dot fluorescence and iSCAT signal was only possible with a longer wavelength scattering beam which is only weakly absorbed by the quantum dot, thereby avoiding excessive blinking and bleaching (660 vs 445 nm). In addition, the imaging speed was lowered to 500 frames/s to ensure <10 nm localization precision in the fluorescence channel while imaging over several seconds. We tracked both the scattering from the gold particle of one head (upper traces in *Figure 3*) and the fluorescence of the quantum dot-labeled head (lower traces in *Figure 3*). In all recorded trajectories of doubly-labeled molecules that exhibited clear transient states in both the fluorescence and scattering channels, the transient states for both heads appeared on the same side of the actin filament (*Figure 3—figure supplement 1*).

## The power stroke is accompanied by a structural transition in the motor domain

Close inspection of the time traces occasionally revealed what appeared to be small backwards movements of the attached head, such as the transition starting at 480 ms in the trace of *Figure 1A*. Since our localisation precision at 1000 frames/s is limited by the intrinsic motion of the label about its attachment point to the motor domain and diffusion of background scatterers, we repeated the tracking assay at 100 frames/s. At this imaging speed, we could frequently observe a transition between two distinct states during each 74 nm step in both an x–y projection (*Figure 4A*) and the time trace (*Figure 4B*). As shown in the lateral trajectory, the transition between these states (AB transition) involves a small, <10 nm, off-axis backward motion of the gold particle attached to the bound head. As a result, the overall step size appears larger than the expected 74 nm step (*Figure 4—figure supplement 1*).

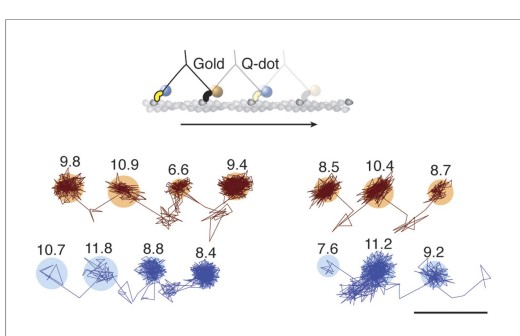

**Figure 3**. Simultaneous scattering and fluorescence tracking of a single myosin 5a. One head was labeled with a 20 nm streptavidin-functionalized gold particle (red) and the other with a fluorescent quantum dot (blue). Reported values correspond to the standard deviation σ in the position of the bound state (nm) and the shaded regions encompass an area of 3σ. The traces are colored according to the label colors in the inset. ATP concentration: 10 μM. Scale bar: 100 nm. Imaging speed: 500 frames/s.

The following figure supplement is available for figure 3:

**Figure supplement 1**. Additional traces of simultaneous scattering and fluorescence tracking of a single myosin 5a.

The positional fluctuations (σ = 0.91 nm SD) of a surface-attached label recorded in the same field of view as the trajectory suggest that we achieved sub-nm lateral localization precision of 20 nm gold at 100 frames/s (*Figure 4B*, blue). The localization noise increased for a gold bead attached to an actin-bound myosin (σ = 1.6 ± 0.3 nm in *Figure 4B*), but remained small enough such that transitions on the order of 5 nm were clearly visible (*Figure 4A–B*). The slightly larger positional fluctuations for gold bound to the actomyosin complex compared to immobilized gold were likely caused by a combination of a flexible protein-label connection and a limited ability in to completely immobilize actin. We observed a clear drop in the iSCAT contrast when the labeled trailing head detached and transitioned to the leading position (*Figure 4C*). We also measured a much smaller change in iSCAT contrast during the AB transition, likely due to a three-dimensional reorientation of the label.

We then performed simultaneous tracking of the head and the tail of myosin 5a, which showed that the small backwards transition corresponds to the power stroke of the bound motor domain. To achieve this, we labeled the myosin head with a 20 nm gold particle as previously and the tail with a GFP booster (*Ries et al., 2012*).

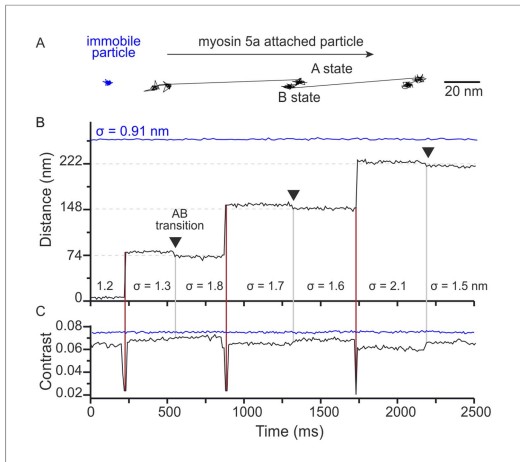

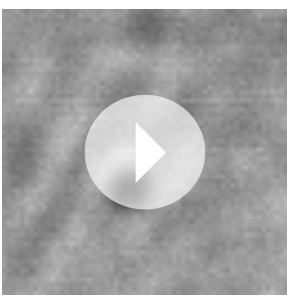

**Video 2.** This file contains a full video from which the trajectory shown in *Figure 4* of the main manuscript was obtained. The video was taken at 100 frames/s and at a magnification of 31.8 nm/pixel. The corresponding field of view shown is $2 \times 2\ \mu m^2$ and corresponds to a total recording time of 5 s. The video playback speed is 100 frames/s.

**Figure 4**. Two states of the motor domain during myosin movement. (**A**) Sample 2D trajectories for a 20 nm gold particle immobilised on the surface and attached to the N-terminus of myosin 5a. The arrow indicates the direction of myosin movement. The trace shown is a fragment of the trajectory obtained from *Video 2*. (**B**) The same trajectory depicted as distance travelled vs time. Two distinct states (**A** and **B**) of the bound head are evident (black arrows). Standard deviations ($\sigma$) are given to compare both fixed (blue) and myosin bound particles (black). (**C**) Corresponding iSCAT contrast time trace for the trajectories in (**B**). A brief reduction in iSCAT contrast coincides with 74 nm steps taken by the labeled head (red vertical lines). The behaviour of a non-specifically surface-bound 20 nm gold particle that is completely immobilized is shown for comparison (blue traces in upper portion of the graphs in panels **B** and **C**). Repeated localization of the particle throughout suggests a nominal sub-nm lateral localization precision and a constant scattering contrast. ATP concentration: 10 $\mu$M. Scale bar: 20 nm. Imaging speed: 100 frames/s.
The following figure supplement is available for figure 4:

**Figure supplement 1**. (**A**) Step size histograms for post to pre-power stroke and post to post power stroke states.

We reduced our frame rate to 20 frames/s in order to increase the fluorescence signal from the tail, since single molecule fluorescence tracking of GFP with <5 nm precision at 100 frames/s is challenging. The resulting traces showed that both the large-scale translocation of the head and the subsequent smaller backwards transition co-incide with translation of the tail (*Figure 5A*). This implies that the small-scale transition accompanies the power stroke of the attached labeled head (AB transition) and is representative of a pre- to post-power stroke transition (*Figure 5B*).

To determine the structural origin of the AB transition, we repeated the tracking experiments with differently sized labels. We found that the average size of the AB transition increased from $7.4 \pm 3.2$ nm to $9.5 \pm 3.1$ nm and $11.5 \pm 2.5$ nm for 20, 30 and 40 nm diameter labels, respectively (*Figure 6A*). This suggests that the AB transition reports on a conformational change of the bound head, which is a rotation of the N-terminal domain, rather than a translation of the head itself along actin. The consequences of a rotational movement of the label are presented schemati-cally in *Figure 6B*. The red circles indicate the positions of two labels in the pre-power stroke A state (with radii $r_1$ and $r_2$, where $r_2 = 2 \times r_1$). The grey circles correspond to the post-power stroke B state. Red and grey dots indicate their respective centers. For a larger label ($r_2$), the same movement of the N-terminus leads to an overall similar motion but with a larger displacement of the center of mass ($d_2$) than for a small label ($d_1$). Using this model and our data, we could extract an angle of rotation, $\alpha$, as well as a distance, $x$, that defines the origin of rotation, since $\sin(\alpha/2) = \frac{1}{2}\, d_1/(r_1 + x)$ and $d_2/d_1 = (r_2 + x)/(r_1 + x)$. Using the hydrodynamic radii for the labels obtained by dynamic light scattering (20, 26 and 32 nm), we obtained $\alpha = 20°$ for the angle and $x = 1.7$ nm.

## Directionality and dynamics of the transient state and the AB transition

Trajectories recorded at 100 frames/s frequently showed both the position of the transient state and the direction of the AB transition, revealing that they were invariably in line (*Figure 7A,B*). During the recording of 351 traces that exhibited clear signatures of the transient state at 100 frames/s,

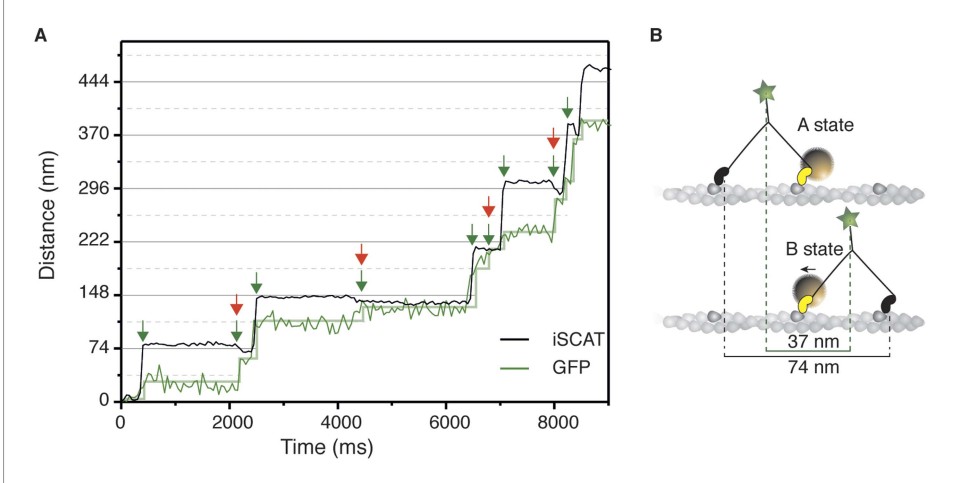

**Figure 5**. Simultaneous iSCAT and fluorescence tracking of myosin 5a. (**A**) Tracking of the scattering signal from the gold nanoparticle attached to the N-terminus (black) and fluorescence signal from the GFP moiety located at the C-terminus of the same myosin 5a molecule (green). Movement of the tail correlates with the labeled head taking its step (74 nm displacement) and with the AB transition (power stroke of the labeled head). Green arrows represent the tail movement, which corresponds to the step of either the labelled or unlabeled head. When the unlabeled head takes its step it coincides with the AB transition within the labeled head (red arrows). Static localization precisions determined as in **Figure 2**: 1.6 nm (iSCAT), 8 nm (GFP). (**B**) Labelling scheme and schematic of the stepping mechanism and the corresponding observables. ATP concentration: 1 μM. Imaging speed for both channels: 20 frames/s.

we found a preference (66% vs 33%) of right over left-handed walking. Within individual traces, we only observed the transient state on the same side of actin. In some cases, however, we could observe individual myosin molecules switch from one actin filament to another. As previously, the transient state always appeared on one side while on one actin filament, but either switched to the other (**Figure 7C**) or remained on the same side (**Figure 7D**) of the actin track when the molecule moved from one filament to another. For 102 switching events, we found that the transient state remained on the same side of the filament 60% of the time, while changing sides in 40% of the events.

Given that the transient state could always be clearly identified either to the right or to the left side of actin, we could align all recorded steps from the 1000 frames/s tracking data by flipping them about the actin filament when necessary to obtain a 2D spatial probability distribution maps of the location of both the unbound and bound myosin head compared to the actin binding sites during a single step (**Figure 8**). The resulting contour plot contained two maxima. For the transient state one was located ~5 nm away from the final actin binding site and the other 40 nm off-axis from the actin filament. For the A and B states the bound head orientations were in line with the position of the transient state, with the lateral projection connecting the A, B and transient states approximately in a straight line (see also **Figure 7**).

Our ability to clearly distinguish between bound and unbound states during processive motion, allowed us to extract the dwell times of the transient state. We found that it followed a single exponential distribution with a lifetime of 17.5 ± 0.6 ms. We found no ATP concentration dependence on both the spatial and temporal distributions of the transient state within our experimental error (**Figure 8—figure supplement 1**). The corresponding dwell time distributions for the A and B states followed the behavior expected for processes limited by ATP binding and ADP release (**Figure 8—figure supplement 2**). In all cases, the dwell times of the A and the B states were identical suggesting that the 20 nm gold particle had no effect on the stepping kinetics.

## Discussion

### N-terminus rotation is associated with lever arm motion

Label-sizes of a few tens of nm are traditionally avoided in single particle tracking experiments to prevent the label from perturbing the dynamics of interest. By monitoring the lifetimes of the A and B states, which effectively provides the dwell times of the labeled and unlabeled heads, respectively, we

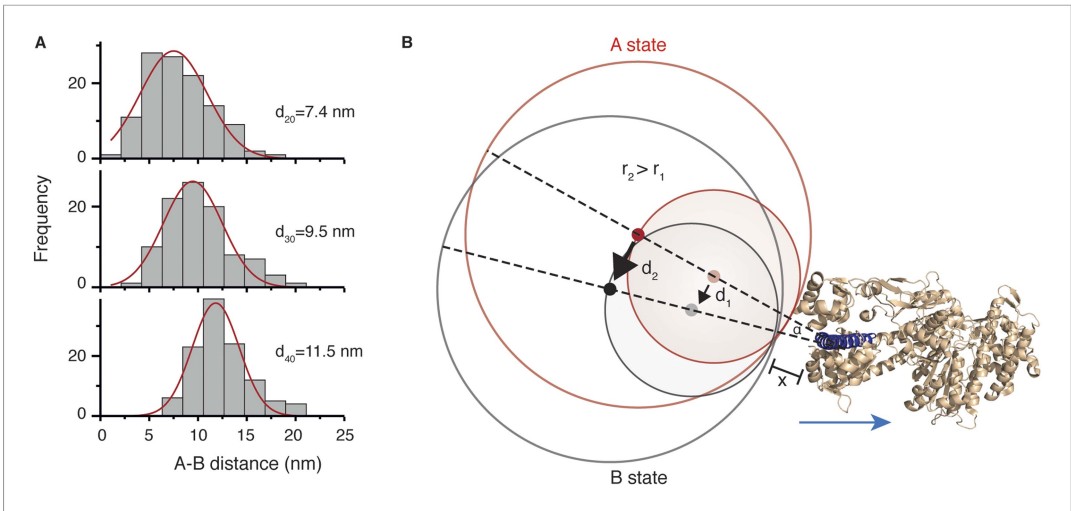

**Figure 6**. Conformational change within the N-terminal domain during the power stroke. (**A**) Histograms for the distances between A and B states for 20, 30 and 40 nm gold nanoparticle labels located at the N-terminus. Total number of steps recorded: 124, 103 and 116, respectively. (**B**) Expected movement for two differently sized labels attached to myosin 5a during a conformational change in the head domain associated with the power stroke. The red circle represents the position of the label in the A state, and the grey circle corresponds to the B state with dots indicating their respective centres of mass. The labels $r_1$ and $r_2$ correspond to the radii of both labels where $r_2 = 2 \times r_1$, and $d_1$ and $d_2$ correspond to the AB distance after rotation by an angle α around an origin located within the head domain at a distance $x$ from the nanoparticle surface. The myosin 5a head domain pre-power stroke conformation is shown in orange (PDB: 1W7J). The lever arm is pointing out (shown in dark blue) and the blue arrow indicates its movement during the power stroke.

demonstrate that the 20 nm gold particle attached to the N-terminus of myosin has no measureable effect on the kinetics of stepping. If detrimental effects can be excluded experimentally, as in this case, our results show that the larger size of the label can actually be advantageous in revealing structural dynamics that would otherwise remain invisible. Here, the label probes the structural change occurring in the bound head during the power stroke and amplifies it into a nanometer scale motion. Previous studies using gold labels were not able to observe this transition likely due to a combination of a different labeling strategy (40 nm or 60 nm gold nanoparticle attached to a calmodulin on the lever arm) and much lower spatial precision (>10 nm) (*Dunn and Spudich, 2007*).

Several EM studies (*Walker et al., 2000*; *Burgess et al., 2002*; *Oke et al., 2010*) failed to detect any obvious conformational differences in the position of the SH3 (N-terminal domain) of lead and trail heads, whereas we now demonstrate a movement of this domain associated with the power stroke taken after the trail head dissociates. This change is consistent with results of molecular dynamic simulations of both myosin 5a and myosin 2 based on crystal structures suggesting that the N-terminus undergoes a rotation during the power stroke transition (*Coureux et al., 2003*; *Cecchini et al., 2008*; *Preller and Holmes, 2013*).

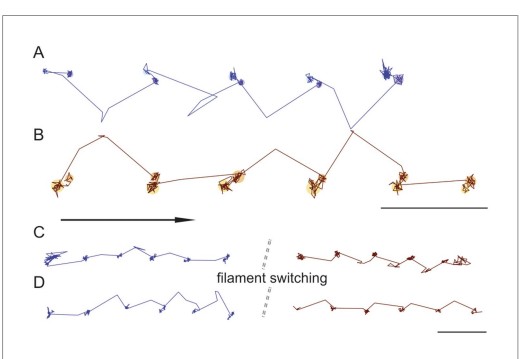

**Figure 7**. Directionality and kinetics of the AB transition and the transient state. (**A** and **B**) Simultaneous observation of the AB transition and the transient state for right and left handed walking molecules. (**C** and **D**) Position of the transient state for the same molecule before and after switching actin tracks. ATP concentration: 10 μM. Scale bar: 100 nm. Imaging speed: 100 frames/s.

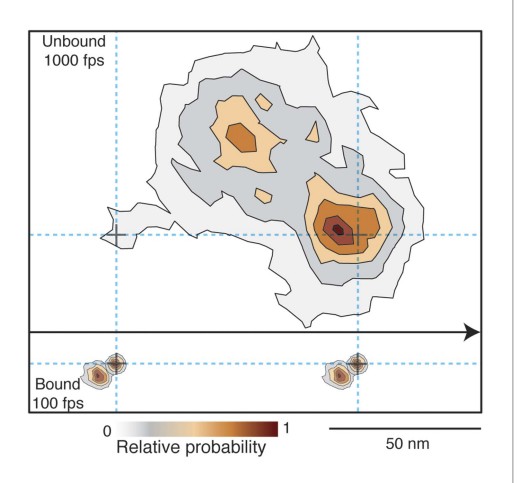

**Figure 8**. Probability density contour maps of the myosin step. Upper panel represents the transient state of the unbound head. Contour map of a two-dimensional histogram with a 10 × 10 nm² bin width obtained from the 1000 frames/s data (N = 486). Lower panel shows the AB transition within the bound head, a two-dimensional histogram with a 1 × 1 nm² bin width generated using the 100 frames/s data (N = 129). All contributing steps were aligned and those to the right of the filament when viewed in the direction of motion were mirrored. The arrow represents direction of movement (from left to right).

The following figure supplements are available for figure 8:

**Figure supplement 1**. Spatiotemporal dynamics of the transient state as a function of ATP concentration.

**Figure supplement 2**. Dwell time distributions for pre and post-power stroke states at different ATP concentrations.

**Figure supplement 3**. Geometrical considerations assuming myosin molecules moving parallel to the glass surface.

**Figure supplement 4**. Probability density contour map of the transient state.

## Sub-steps along actin and leading head detachment are not significant contributors to the mechanochemical cycle of myosin 5a

Although the original myosin 5 tracking performed with a fluorescently-labeled calmodulin bound to the lever arm at 1.5 nm precision revealed 74 nm steps, (*Yildiz et al., 2003*) a later study with rigidly attached dye labels on the calmodulin bound to the lever arm suggested that they in fact represent 64–10 nm steps (*Syed et al., 2006*). During the acquisition of 4728 steps in 635 trajectories, we never observed a large followed by a small forward step. In our work, we labeled the motor domain directly, while in the previous studies, calmodulin bound to the lever arm was used as an attachment point. Since the lever arm moves significantly during the power stroke, attachment of the dye at this location may suggest a sub-step along actin, even though the motor domain itself does not move. Thus the term 'sub step' is inappropriate for this pattern of movement. In addition, the orientation of single dipoles can have a significant impact on the position obtained from standard Gaussian fitting and exacerbate even very small translocations (*Enderlein et al., 2006*).

A recent AFM study (*Kodera et al., 2010*) reported that detachment and reattachment of the leading head, a so-called 'foot stomp' contributes to the mechanochemical cycle of myosin 5. Such behavior was also observed in the single molecule tracking study discussed above, (*Syed et al., 2006*) although the frequency of occurrence was not discussed in that work. The sensitivity of our tracking methodology to detachment from actin in three dimensions, however, shows that myosin remains firmly bound to actin with both heads during the mechanochemical cycle irrespective of whether the head is leading or trailing (*Figure 4—figure supplement 1*), except when the head takes a 74 nm step. Only on very rare occasions did we observe detachment of the leading head. Specifically, only ~3% of steps taken at 1 µM ATP, ~0.6% of steps taken at 10 µM ATP, none at saturating ATP exhibited this behavior.

The unbinding event was not necessarily followed by reattachment at a different site suggesting that these events can be explained simply by the binding equilibria between myosin and actin, rather than being an active component of the stepping mechanism. One likely explanation for the increased occurrence of lead head detachment and reattachment in recent AFM studies (*Kodera et al., 2010*; *Ando et al., 2013*) is the much lower ATP concentration used or the non-negligible interaction of the AFM tip with the protein.

## Structurally constrained diffusion leads to unidirectional motion of myosin 5a

The combination of temporal and spatial precision achieved in this work directly reveals the motion of the unbound head of myosin during translocation. From the two head bound state of myosin 5a, the

trailing head detaches from actin upon binding of ATP (*Figure 9A*). The leading head then undergoes a power stroke that leans the protein forward, a motion that exhibits a strong torsional component (*Komori et al., 2007*; *Ohmachi et al., 2012*). After the power stroke, the unbound head arrives at a minimum in the potential energy from where it approaches the next actin binding site in what appears to be a one dimensional biased search. The transient state lifetime agrees with the duration of increased flexibility reported in optical trapping experiments probing the attachment stiffness at high time-resolution (*Veigel et al., 2002*) and single molecule tracking of the motor domain (*Dunn and Spudich, 2007*; *Beausang et al., 2013*). The measured lifetime also correlates well with previously reported ensemble rate constant for the weak-to-strong binding transition (47 s$^{-1}$) (*Rosenfeld and Sweeney, 2004*). The presence of an additional high probability density very close to the final binding site further favors such an interpretation. This density very likely corresponds to the weakly bound state from which the head frequently moves back to the side position.

An important implication of our data is that the position of the transient state, displaced 40 nm along and perpendicular to the filament is inconsistent with a purely rotational diffusion search. In that case we would require most molecules to be bound parallel to the surface and then expect the average position of the detached head to be displaced ~56 nm along and not more than 20 nm perpendicular to the filament (*Figure 8—figure supplement 3*). We emphasize however, that our data does not contradict previous dark field results, which lead to the rotational diffusion model (*Dunn and Spudich, 2007*). We can closely reproduce such results by adding 17 nm of localization noise to our traces. At such localization precision, left- and right- handed walking is no longer distinguishable leading to a probability density map that is no longer one-sided (*Figure 8—figure supplement 4*).

Why the unattached head occupies a positon 40 nm away from the actin filament is a priori puzzling. There is neither an intrinsic reason for the head to pause there, nor is there a binding site present at this position. Our experimental data provides no evidence that the transient state is induced or measurably affected by unwanted interactions caused by the particle or the surface as suggested by the similarity of our transient state lifetime to previous studies with alternative labeling strategies, (*Forkey et al., 2003*; *Snyder et al., 2004*; *Warshaw et al., 2005*) the presence of the transient state for much smaller labels (*Figure 1—figure supplement 2*), the change in contrast during the step (*Figure 2*), the lack of non-specific binding of our label to the surface or actin and any localizations during the one head bound state that coincide with the underlying actin track. This location and the resulting movement, however, become more intuitive when considering that the intrinsic angle between the lever arms is similar to that found when the two heads are bound to actin, that is, ~37 nm away from each other. Comparison of electron micrographs of myosin 5 bound to actin (*Walker et al., 2000*) and non-specifically bound to a surface in the

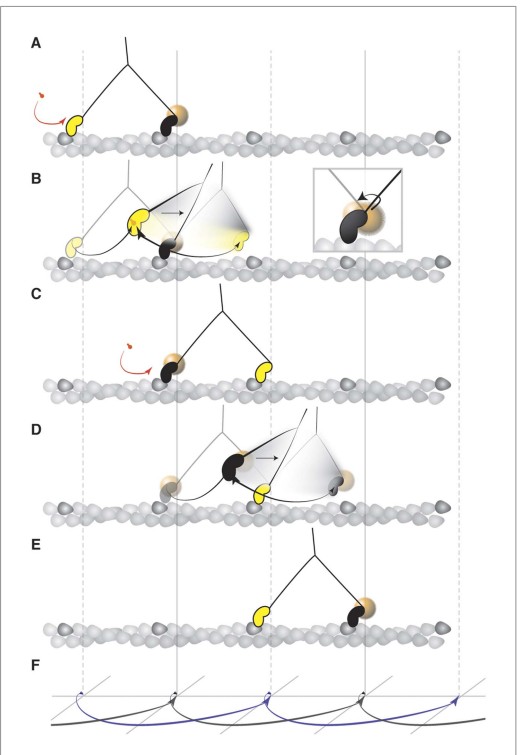

**Figure 9**. Mechanism of the myosin step. (**A**) ATP binding to the trailing head (yellow) and its detachment releases strain stored in the molecule. (**B**) The bound head (black with a gold nanoparticle attached) performs its power stroke which is accompanied by the AB transition (inset). The labeled head becomes a new trailing head (**C**) which detaches after ATP binding. (**D**) It moves forward in a partially twisting motion and occupies an off axis position (transient state). From this position, the head binds the desired binding site while ATP hydrolyses. (**E**) The step completes as the head binds actin and is repeated by the other head in the same direction dictated by the initial torsional strain. (**F**) Schematic representation of both heads' movement (lateral projection).

absence of actin (*Takagi et al., 2014*) show clearly peaked angle distributions around 105 and 115°, respectively, suggesting a built-in preference for such a spatial arrangement of the two heads, rather than a truly flexible linkage, another indication that a completely free swivel at the neck-linker is unlikely. In this way, keeping the angle between the lever arms roughly intact during stepping and twisting partially about the leading head causes the translocating motor domain to preferentially reach the desired binding site. As a result a full three dimensional search mechanism is minimized to one defined by a one dimensional arc.

The position of the transient state together with what appears to be constrained diffusion suggests that myosin side steps along actin in a combined twisting and leaning motion, much more like a drawing compass rather than a freely jointed swivel or human-like gait. In hindsight, such a mechanism has numerous advantages over a search based on full rotational diffusion. It generates a built-in bias for finding the correct binding site and ensures that the head needs to travel and search only along one dimension to find the desired binding site defined by the torsion about the leading head and neck linker rather than a lower probability three-dimensional Brownian search. Even if the motor domain fails to bind strongly on first passage, it swings back and forth between the transient state and the desired actin site until tight binding can occur. This suggests that the structure of myosin 5a facilitates and controls the motion of the unbound head to achieve such high specificity in finding the desired binding site.

Our results do not imply that the head is effectively stationary at the transient state position while in the one head bound state. Some degree of diffusional search is inherent to the system and cannot be captured even at our low exposure time of 0.56 ms, which is comparable to the positional autocorrelation time we measured (0.3 ms), and predicted theoretically (*Craig and Linke, 2009*). The location of the transient state, the fact that in several thousand steps we never observed a single step with transient localizations on both sides of the actin filament, and a predominantly perpendicular orientation of myosin relative to the surface caused by our surface preparation, however, suggest that diffusion is constrained. On the contrary, we believe that this constraint is a key to myosin 5's efficiency in terms of processive motion in contrast to a completely free, spherical search space.

## Relationship between the transient state and the AB transition

The fact that the AB transition and the relative location of the rear-binding site and the transient state are in line has two further important implications. It suggests that the AB transition cannot solely be a consequence of a steric interaction between the label and the lever arm. In that case, we should only observe one type of AB transition, irrespective of whether right or left-handed walking occurs. Furthermore, it shows a connection between the directionality of the N-terminus and lever arm movement. The observed N-terminus domain movement could thus be a signature of the release of this strain. This conformational change, critical for rearrangement of the nucleotide binding pocket and/or closure of the internal cleft, would then allow for finalizing the actomyosin energy transduction. In other words, the energy from the ATPase reaction, stored in the actomyosin complex can only be fully used after the trailing head detaches and the system can relax.

Our data shows that the AB transition is associated with the power stroke but it is unlikely that the 20° rotation of the N-terminus is directly responsible for the side step per se. The fact that the transient state is either to the left or to the right of the filament implies that it cannot be a pure consequence of the attached lever arm swing. We therefore propose that the position of the transient state and the direction of the AB transition reflect the strain release upon rear head detachment. The initial strain is built up within the head domain and transmitted up to the lever arm during the first binding when one of the heads twists in order to bind to the same actin filament (*Liu et al., 2006*).

The simultaneous tracking of the movement of both heads using the combination of gold scattering and fluorescence demonstrates that myosin walks in a symmetric manner where each head movement is accompanied by an 180° swing on the same side of the actin filament, unlike a human gait. This type of movement would imply that the cargo also rotates with each step or that a swivel exists to relieve the strain that would otherwise develop between the myosin and the cargo. Such a rotation has been measured by the orientation of quantum rods attached to the myosin 5a tail, although, in this work the source of the rotational movement was thought to be thermal (*Ohmachi et al., 2012*).

Our observation that the orientation of the transient state relative to actin may change for the same molecule by binding to another actin filament shows that the position of the transient state relative to actin is not an intrinsic property of individual myosin molecules. Otherwise, we would expect each molecule to preserve the orientation of the transient state even when changing from one filament to

another. Instead, the decision is made upon binding to actin, with both cases following the right and left hand preference previously mentioned. With respect to the source of the symmetry breaking, one possible explanation involves the steric interaction with the surface. Depending on the initial binding angle the steric interactions with the surface would determine the direction of rotation. However, if the sidedness was only a consequence of an interaction with the surface, then we would at least expect the molecules that are bound perpendicular to the glass surface and thus have the least interaction with it, to show no preference. Since we never observed transient states on both sides of the filament during individual runs, the reported mechanism is likely to be an intrinsic feature of the myosin 5a stepping.

Based on our observation that the transient state for both heads occurs on the same side of the actin filament we propose that the twisting that takes place during the initial binding event of both heads to actin determines the directionality. This initial twist, possibly driven by thermal fluctuations, establishes the direction in which the initial torsional strain is built up and is then repeatedly stored and released as the protein moves along actin. Once the directionality of the transient state has been established, both heads follow the same constrained route resulting in an efficient, unidirectional motion.

## Materials and methods

### Sample preparation

Rabbit skeletal muscle actin was prepared as described (*Spudich and Watt, 1971*) and stored in liquid nitrogen until used. A 20 µM actin stock solution was prepared in polymerisation buffer (10 mM imidazole, 50 mM KCl, 1 mM $MgCl_2$, 1 mM EGTA [pH 7.3] containing 1.7 mM DTT, 3 mM ATP). Actin was diluted in motility buffer (MB; 20 mM MOPS pH 7.3, 5 mM $MgCl_2$, 0.1 mM EGTA) 20–50 times. Mouse myosin 5a HMM with a C-terminal GFP was expressed in the presence of calmodulin and purified as described (*Wang, 2000*). In addition, the N-terminus was modified by the addition of a nucleotide sequence encoding an AviTag peptide (GLNDIFEAQKIEWHE) for site-specific biotinylation with BirA ligase (Avidity, Aurora, Colorado). After biotinylation, the sample was aliquoted, flash frozen in 20 µl drops and stored at −80°C. Before labeling, the myosin sample was diluted in MB containing 40 mM KCl, 5 mM DTT, 0.1 mg/ml BSA and 5 µM calmodulin.

Gold nanoparticles of 20 nm, 30 nm and 40 nm in diameter conjugated with streptavidin were purchased from BBI (UK) and directly mixed with biotinylated myosin 5a sample in a 4:1, gold to myosin ratio, consistent with one or zero myosin molecules per gold particle. The mixture was incubated on ice for at least 15 min (sample volume 50 µl, final concentration of myosin 300 pM). The same procedure was used for double gold/Qdot labeling. The quantum dots (emission maximum 565) conjugated with streptavidin were purchased from Invitrogen (Lifetechnologies, UK).

For fluorescence only imaging, myosin was incubated for 10 min with Atto-647 streptavidin (Atto-TEC, Germany). The following oxygen scavenger system was used to increase the fluorescent dye stability: 0.2 mg/ml glucose oxidase, 0.4% wt/vol glucose, 0.04 mg/ml catalase, all purchased from Sigma-Aldrich (UK).

The flow cell was prepared as described (*Dunn and Spudich, 2011*). It was first rinsed with 1 mg/ml solution of poly(ethylene glycol)-poly-l-lysine (PEG-PLL) branch copolymer (Surface Solutions SuSoS, Switzerland) in PBS and incubated for 30 min. Next, it was washed twice with MB before adding the actin solution. After 5 min of incubation, the chamber was washed with MB and the surface was blocked by adding 1 mg/ml BSA in MB buffer. Finally, the chamber was inspected and myosin-gold conjugate solution containing ATP was added. All assays were performed at room temperature.

### Experimental setup

Interferometric scattering microscopy was performed as detailed previously (*Ortega-Arroyo and Kukura, 2012*). For two-colour imaging a second diode laser (635 nm) was overlapped with the 445 nm beam path with a dichroic mirror. In the detection arm, the images were separated by an identical optic before being imaged onto two separate CMOS cameras (MV-D1024-160-CL-8, Photonfocus, Switzerland) at 333× magnification (31.8 nm/pixel). The incident power was adjusted to 17.9 kW/cm$^2$ at the sample to achieve near-saturation of the CMOS camera, which ensured shot noise limited detection. Fluorescence only imaging and tracking was achieved with a home-built TIRF microscope using a 635 nm diode laser. The incident power was set to 5 kW/cm$^2$ and the fluorescence imaged onto an Andor iXon3 860 EM-CCD camera at 72.1 nm/pixel with a 9.2 µm × 9.2 µm field of view using dielectric filters (Thorlabs, Germany) to separate illumination and emission.

## Image processing

Sample-specific images with shot noise limited sensitivity were obtained by dividing the raw image by a reference flat field image. The flat field reference image containing non-sample specific illumination inhomogeneities, fixed pattern noise and any constant background caused by residual reflections was produced by first recording 2000 images while moving the sample stage with a piezo-driven XY translation stage. The resulting image stack was temporally averaged to 100 frames and each pixel of the flat field reference image was computed as the temporal median value of the frame sequence. Removal of sample specific constant background and thereby increase in the image signal to noise ratio was achieved by obtaining the static background from the sample by averaging at least 10 frames that lacked a signal from the processive myosin 5a molecule labeled with a gold nanoparticle of interest (*Ortega Arroyo et al., 2014*). Nanometric tracking of labeled myosin 5a was achieved by non-linear least square fitting of the point spread function to a two-dimensional Gaussian.

## Data analysis

Spatiotemporal characterization of the transient state was achieved by alignment of several steps and separation of the states. We segmented each trajectory into step pairs defined as the set containing the data before and after each 74 nm step. These steps were detected manually by using a time interval slider along the 2D trajectory and checking for changes in positional fluctuations. The orientation of the transient state relative to actin was determined and later used to align all steps.

To determine the spatiotemporal dynamics of the side step only step pair traces that showed transient states lasting longer than 10 ms were used. First, we fixed the center of mass of the beginning of the step to coordinate position (0,0). Then, we applied a rotation matrix to the entire step pair trace so that both bound states lie along the horizontal axis. Finally, all aligned step pair traces were overlapped to produce the contour plots shown in *Figure 8* and *Figure 8—figure supplement 1*.

## Acknowledgements

We would like to thank Dr Attila Nagy for assistance on the myosin 5a construct design, Dr Katelyn Spillane for initial assistance with the myosin construct and Dr Christof Gebhardt for helpful discussions. JOA was supported by a scholarship from CONACyT (scholar: 213546), JA by a Marie Curie Fellowship (330215), JRS by the intramural funds from the National Heart, Lung, and Blood Institute, National Institutes of Health (ZIA HL004229), and PK by the John Fell Fund, a career acceleration fellowship by the EPSRC (EP/H003541) and an ERC starting grant (NanoScope).

## Additional information

### Funding

| Funder | Grant reference | Author |
| --- | --- | --- |
| European Research Council (ERC) | Nanoscope | Philipp Kukura |
| Engineering and Physical Sciences Research Council (EPSRC) | EP/H003541 | Philipp Kukura |
| European Research Council (ERC) | 330215 | Joanna Andrecka |
| National Heart, Lung, and Blood Institute (NHBLI) | ZIA HL004229 | James R Sellers |
| Consejo Nacional de Ciencia y Tecnología | scholar: 213546 | Jaime Ortega Arroyo |

The funders had no role in study design, data collection and interpretation, or the decision to submit the work for publication.

### Author contributions

JA, JOA, Acquisition of data, Analysis and interpretation of data, Drafting or revising the article; YT, JRS, Drafting or revising the article, Contributed unpublished essential data or reagents; GW,

Acquisition of data, Analysis and interpretation of data; AF, Stalk-labeling of myosin 5, Analysis and interpretation of data; LMK, Statistics of single molecule filament switching events, Analysis and interpretation of data; GY, Statistics of single step behaviour, Analysis and interpretation of data; PK, Conception and design, Drafting or revising the article

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
