## [Decision Letter]

Thank you for sending your work entitled “Structural dynamics of myosin 5 during processive motion revealed by interferometric scattering microscopy” for consideration at *eLife*. Your article has been favorably evaluated by John Kuriyan (Senior editor) and 2 reviewers, one of whom is a member of our Board of Reviewing Editors.

The Reviewing editor and the other reviewer discussed their comments before we reached this decision, and the Reviewing editor has assembled the following comments to help you prepare a revised submission.

In this manuscript, Kukura and colleagues use interferometric scattering microscopy termed iSCAT to observe the movement of a 20 nm gold particle attached to a myosin V head with 1 ms time resolution and 5 nm spatial precision. Interestingly, the labeled head visits a transiently detached state located about 40 nm off axis, and it alternates between a forward on-axis position and the off-axis position a few times before settling on the new forward on-axis position. Remarkably, the sense (right vs left handed) of this off-axis intermediate is the same for both heads (as they demonstrate using correlated imaging of the other head labeled with a quantum dot at 10 ms time resolution), supporting the symmetric hand-over-hand mechanism. The resulting model posits that the dimeric motor moves like a compass, rotating around its bound head during the powerstroke, and the direction of this rotation is the same through a processive period. But whether the rotation is clockwise or counter-clockwise appears to be determined stochastically upon initiation of the movement. They also combined this with fluorescence particle tracking of quantum dots or GFP to correlate motions of the gold-labeled heads with the partner head or myosin tail. The data also suggest that there is no-substep per se and that stomping of the front head, as was proposed based on AFM images and an earlier fluorescence study, may not be an essential part of the stepping mechanism. Additional data on the slight movement of the bound head upon powerstroke support the conformational change of the bound head instead of it staying in the same conformation. Overall, the work is novel and provides new insights and an overall simple and clean model on myosin V motility. They also conclude that the search by a detached head for the next actin binding site does not sweep out the sphere expected from a free swivel at the head-tail junction, but rather sweeps along an arc limited by a relatively fixed angle between the two lever arms. It's excellent work, carefully executed, and the paper is generally written very clearly. The paper will be of interest to investigators in the motor field and also for tracking microscopists more broadly. Several mechanisms alternative to the ones proposed are not discussed, but otherwise, the paper is outstanding.

Major comments:

1) What evidence exists to rule out the idea that the molecules are mostly walking along the right side of the actin filament limited by the glass, or the left side, again constrained by the glass slide? Then the transient state could be approximately in the plane of the slide on one side or the other. This would look sideways to the camera from below, but it's not sideways from the molecule's perspective. The predominance of left-side walking could be due to the native weak helical pitch of myosin's path. This would also explain why the sidedness would switch when the myosin finds a different actin filament. If there is reason to rule out this simple explanation of the main result, then that should be discussed. If the actin were also visualized, then, experimental evidence could be given.

2) It is not clear what the source of the symmetry breaking, that is, whether the motor rotates counter-clockwise or clockwise. Given that the actin filament is attached to the surface, interactions with the surface or constraints provided by the surface are expected to play a significant role but this possibility was not mentioned. Why? Even if the symmetry breaking is an artifact of the experimental configuration, it should be fine as long as the work reveals an intrinsic feature of the walking mechanism.

3) AB transitions occur along the line drawn to the off-axis intermediate. Is this a pure coincidence? I would suppose so because the angle of head rotation, 20 degrees, is clearly very different from the much larger rotation angle for the detached head that undergoes during the compass-like movement.

4) The idea that the search space is the full spherical range of the detached head on a freely swiveling head-tail junction is dismissed because the transient position is well localized. However, if the diffusion rate is much faster than the frame time, even at 1000 frames/s, then the signal would give the center position and the spot would not noticeably broaden. Again, discuss this or the evidence against. The two previous studies of dynamics during the search (Dunn and Spudich, 2007, and Beausang et al., 2013) were both consistent with the full spherical search space. Is the presently postulated one-dimensional arc consistent with those earlier studies?

5) The constrained angle between the attached and detached heads, at 37 nm, would imply that there is no intramolecular strain between the heads upon attachment. But this strain is necessary to distinguish the ADP off-rate between the leading and trailing heads to obtain directionality. The authors might argue that the intermolecular strain is only developed once Pi is released. If that is the idea, then so state. There would be a kinetic issue with detachment of the trailing head before the strain is imposed that might limit processivity.

---

## [Author Response]

*1) What evidence exists to rule out the idea that the molecules are mostly walking along the right side of the actin filament limited by the glass, or the left side, again constrained by the glass slide? Then the transient state could be approximately in the plane of the slide on one side or the other. This would look sideways to the camera from below, but it's not sideways from the molecule's perspective. The predominance of left-side walking could be due to the native weak helical pitch of myosin's path. This would also explain why the sidedness would switch when the myosin finds a different actin filament. If there is reason to rule out this simple explanation of the main result, then that should be discussed. If the actin were also visualized, then, experimental evidence could be given*.

We agree with the referees that it is in principle possible that molecules bound parallel to the glass surface and under the assumption of rotational diffusion would exhibit an average position of the detached head to the side of the actin filament (Figure 10).

Author response image 1.**DOI:**
http://dx.doi.org/10.7554/eLife.05413.026

Such a scenario, however, is inconsistent with a number of experimental observations:

a) Geometrical considerations: If the head searched the full spherical space, then the average position would be simply about half way between the leading head and the next binding site (see Figure 10, purple). For a parallel orientation this would be 56 nm forward and 15 nm off axis and for a perpendicular orientation 56 nm forward but “in line” with the bound head positions. These numbers arise from simple geometric considerations but also agree with much more sophisticated simulations based on a free rotational diffusion model (Hinczewski et al., 2013, PNAS). We, however, observed a transient state half way between the two binding sites, 40 nm off axis (marked in orange). Given the head-head separation of 37 nm, our observations are more consistent with myosin maintaining the angle between lever arm. These considerations are discussed in the subsection headed “Structurally constrained diffusion leads to unidirectional motion of myosin 5a” of the manuscript.

b) Our assay makes sideways binding rather unlikely, in contrast to the assay used by Ando and coworkers for example. Incubation with BSA after actin binding blocks the surface making the binding sites on the side of actin rather inaccessible in contrast to those at the top. In addition, perpendicular binding will be likely preferred as it minimizes interaction of the label with the surface.

c) In order to verify that most of our molecules indeed bind perpendicular to the glass we labeled the (∼27 nm long) tail of our myosin construct and took advantage of the resulting myosin 5a height (∼43 nm). We attached streptavidin functionalized, 20 nm gold particles using biotinylated antibody against GFP. In this assay, all gold labels produced a positive iSCAT contrast when placed into focus (bright), in contrast to head-labeled gold, which invariably appeared dark (see Figure 11). We remark that the iSCAT contrast of gold particles bound to a surface in water cannot be inverted by simply changing the focusing but requires an additional phase contribution from a difference in optical pathlength between scattered and reflected light (Ortega-Arroyo and Kukura, 2012, Phys Chem Chem Phys). Although precise z-calibration was difficult for reasons already discussed in the text, these observations strongly suggest that gold particles attached to the tail are 40–60 nm above those bound to the head in most cases.

Author response image 2.**DOI:**
http://dx.doi.org/10.7554/eLife.05413.027

The figure containing raw iSCAT images illustrates the point made above: head and tail labeled myosins (see insets) imaged without changing the focus on the same sample region. (A) Head labeled myosin appears dark as marked with the black arrows. (B) After washing away the sample by flushing with the motility buffer, tail labeled myosin was loaded into the chamber appearing bright (white arrows). Nonspecifically bound particles remained visible and exhibited negative contrast (red crosses).

d) Another implication of the parallel-bound model is that molecules bound with both parallel and perpendicular orientations would exhibit roughly the same contrast change when the head detaches. For parallel bound molecules, volume exclusion with the surface would result in an average height of about 15 nm above actin. For perpendicular bound molecules, the average position of a freely rotating head would be located slightly above the lever arms’ junction. As we show in Figure 2—figure supplement 1, we occasionally encounter tracks where the transient state cannot be localized with blue illumination because the iSCAT contrast drops to zero. This is only possible if the particle travels ∼40 nm in along the optical axis, which is inconsistent with the scenario suggested by the referees. This interpretation is strengthened by simultaneous iSCAT measurements using blue (445 nm) and red (635 nm) illumination. In that case, the particle remained visible in the red channel while it disappeared in the blue. At the same time, the off-axis component of the transient state is much smaller compared to those trajectories when the transient state remains visible in both channels, which is most consistent with a parallel bound molecule lifting the detached head ∼40 nm up (Figure 2—figure supplement 1). We have not referred properly to this figure in the original version of the manuscript, but have now corrected this mistake and added a short discussion in the subsection “Revealing the unbound head motion in three dimensions”.

*2) It is not clear what the source of the symmetry breaking, that is, whether the motor rotates counter-clockwise or clockwise. Given that the actin filament is attached to the surface, interactions with the surface or constraints provided by the surface are expected to play a significant role but this possibility was not mentioned. Why? Even if the symmetry breaking is an artifact of the experimental configuration, it should be fine as long as the work reveals an intrinsic feature of the walking mechanism*.

It is indeed possible that the initial binding angle determines the direction of rotation through a steric interaction with the surface. The reported mechanism, however, is likely to be an intrinsic feature, since we never observed transient states on both sides of the filament during individual runs. If the sidedness was only a consequence of an interaction with the surface, then we would at least expect the molecules that are bound perpendicular to the glass surface and thus have the least interaction with it, to show no preference.

We have added a brief discussion of this aspect in the subsection “Relationship between the transient state and the AB transition” of the manuscript.

*3) AB transitions occur along the line drawn to the off-axis intermediate. Is this a pure coincidence? I would suppose so because the angle of head rotation, 20 degrees, is clearly very different from the much larger rotation angle for the detached head that undergoes during the compass-like movement*.

We have no experimental data that would suggest that this is not a coincidence. We remark, however, that the conformational change in the head is not responsible for the side step per se. What our data shows is that the direction of the conformational change is in line with the position of the transient state, which suggest that the AB transition is a consequence of the strain release and the direction depends on the initial strain build up.

We have clarified this in the subsection headed “Relationship between the transient state and the AB transition” together with the response to comment 5.

4) The idea that the search space is the full spherical range of the detached head on a freely swiveling head-tail junction is dismissed because the transient position is well localized. However, if the diffusion rate is much faster than the frame time, even at 1000 frames/s, then the signal would give the center position and the spot would not noticeably broaden. Again, discuss this or the evidence against. The two previous studies of dynamics during the search (Dunn and Spudich, 2007, and Beausang et al., 2013) were both consistent with the full spherical search space. Is the presently postulated one-dimensional arc consistent with those earlier studies?

We point the referees to our response to comment 1, which partially addresses this point as it shows that our assay predominantly results in molecules aligned roughly perpendicular to the surface and that the position of the transient state is inconsistent with a fully spherical search. In addition, our results contain several pieces of evidence contrary to such behavior, detailed below and now discussed in the subsection headed “Structurally constrained diffusion leads to unidirectional motion of myosin 5a”.

a) A freely diffusing head would be represented by only one average position. Our data, however, show two clear maxima, one near the desired binding site and one representing what we termed the transient state. How confined the motion is near these two points is indeed difficult to determine. The similarity in the spatial extents of the two maxima in the probability density maps suggests that confinement may be surprisingly tight. In fact, it is difficult to envision how the transient state to the side of actin and that near the binding site could exhibit the same spatial extent in the probability density if the latter is highly localized (must be as the molecule is extended) and the former is a result of a completely free spherical search.

b) In order to quantify the diffusion rate of the unbound labeled head, we repeated experiments at much faster imaging speed of 10,000 frames/s. The time traces of transients states lasting longer than 5 ms from 10 trajectories were selected for further analysis (N = 37, Figure 12). The autocorrelation function for each isolated position time trace was calculated and later averaged to produce the ensemble autocorrelation function for the transient state. The diffusive time (τ_d_ =0.30 ms, in good agreement with theoretical calculations; Craig and Linke, 2009, PNAS) was extracted by fitting the autocorrelation function to a mono-exponential.

Author response image 3.**DOI:**
http://dx.doi.org/10.7554/eLife.05413.028

This result shows that the diffusion rate is comparable to the exposure time of 0.56 ms. For complete averaging of the position one would indeed require a much faster diffusion than the frame time, which is not the case here. We cannot rule out, however, that the diffusional search space is restricted by the label itself. If there were a strong dependence on label size, however, it is surprising that experiments with much smaller labels (atto-647N) revealed transient states in a similar location (see Figure 1—figure supplement 2).

We now explicitly discuss these data and the option and implications of a free diffusional search in the subsection “Structurally constrained diffusion leads to unidirectional motion of myosin 5a”. It is without question that some positional averaging occurs on the time scale of our experiment*.* We believe, however, that the bulk of our observations remain inconsistent with a completely free spherical search space.

To which degree Dunn and Spudich, 2007, and Beausang et al., 2013 are in fact consistent only with a free spherical search, we believe, is debatable. Regarding Dunn and Spudich, 2007, we can almost perfectly reproduce the previous results assuming 18 nm of localization noise. At this stage, left- and right-handed walking is no longer distinguishable leading to a probability density map that is no longer one-sided. We have now included this figure as Figure 8—figure supplement 4 and a short discussion in the aforementioned subsection of the Results. Similarly, our data also agrees very well with an increased mobility of the translocating head measured by single molecule fluorescence polarization (Beausang et al., 2013, Biophys J). The effective time-resolution of this experiment was 0.8 ms after binning to collect sufficient photons, longer than the exposure time used here (0.54 ms). If the stepping head indeed was on a freely jointed swivel and explored the full rotational space on a much faster timescale, these experiments would not have detected the reported wobble, but instead only an average dye orientation, which was different from that when the head is bound to actin, either leading or trailing. The fact that Beausang et al., 2013 observed a wobble is thus in excellent agreement with our observation of constrained diffusion, as is our transit time of 17 ms compared to 13 ms.

*5) The constrained angle between the attached and detached heads, at 37 nm, would imply that there is no intramolecular strain between the heads upon attachment. But this strain is necessary to distinguish the ADP off-rate between the leading and trailing heads to obtain directionality. The authors might argue that the intermolecular strain is only developed once Pi is released. If that is the idea, then so state. There would be a kinetic issue with detachment of the trailing head before the strain is imposed that might limit processivity*.

The implication of no intermolecular strain would only follow if the source of the strain is exclusively located at the lever arm. We believe, however, that a significant source of the strain is located in the motor domain and can be transmitted, e.g. through the converter, up to the lever arm. This interpretation comes from the largely torsional nature of the step revealed in our study.

We now added a paragraph to clarify this in the subsection headed “Relationship between the transient state and the AB transition”.